# Systematic review and meta-analysis of the prevalence of common respiratory viruses in children < 2 years with bronchiolitis in the pre-COVID-19 pandemic era

Sebastien Kenmoe[1], Cyprien Kengne-Nde[2], Jean Thierry Ebogo-Belobo[3], Donatien Serge Mbaga[4], Abdou Fatawou Modiyinji[1,5], Richard Njouom[1]*

1 Department of Virology, Centre Pasteur of Cameroon, Yaoundé, Cameroon, 2 National AIDS Control Committee, Epidemiological Surveillance, Evaluation and Research Unit, Yaounde, Cameroon, 3 Medical Research Centre, Institute of Medical Research and Medicinal Plants Studies, Yaoundé, Cameroon, 4 Department of Microbiology, Faculty of Science, The University of Yaounde I, Yaoundé, Cameroon, 5 Department of Animals Biology and Physiology, Faculty of Sciences, University of Yaoundé I, Yaoundé, Cameroon

* njouom@pasteur-yaounde.org, njouom@yahoo.com

## Abstract

### Introduction

The advent of genome amplification assays has allowed description of new respiratory viruses and to reconsider the role played by certain respiratory viruses in bronchiolitis. This systematic review and meta-analysis was initiated to clarify the prevalence of respiratory viruses in children with bronchiolitis in the pre-COVID-19 pandemic era.

### Methods

We performed an electronic search through Pubmed and Global Index Medicus databases. We included observational studies reporting the detection rate of common respiratory viruses in children with bronchiolitis using molecular assays. Data was extracted and the quality of the included articles was assessed. We conducted sensitivity, subgroups, publication bias, and heterogeneity analyses using a random effect model.

### Results

The final meta-analysis included 51 studies. Human respiratory syncytial virus (HRSV) was largely the most commonly detected virus 59.2%; 95% CI [54.7; 63.6]). The second predominant virus was Rhinovirus (RV) 19.3%; 95% CI [16.7; 22.0]) followed by Human bocavirus (HBoV) 8.2%; 95% CI [5.7; 11.2]). Other reported viruses included Human Adenovirus (HAdV) 6.1%; 95% CI [4.4; 8.0]), Human Metapneumovirus (HMPV) 5.4%; 95% CI [4.4; 6.4]), Human Parainfluenzavirus (HPIV) 5.4%; 95% CI [3.8; 7.3]), Influenza 3.2%; 95% CI [2.2; 4.3], Human Coronavirus (HCoV) 2.9%; 95% CI [2.0; 4.0]), and Enterovirus (EV) 2.9%; 95% CI [1.6; 4.5]). HRSV was the predominant virus involved in multiple detection and most

**Data Availability Statement:** All relevant data are within the manuscript and its Supporting Information files.

**Funding:** The authors received no specific funding for this work.

**Competing interests:** The authors have declared that no competing interests exist.

codetections were HRSV + RV 7.1%, 95% CI [4.6; 9.9]) and HRSV + HBoV 4.5%, 95% CI [2.4; 7.3]).

## Conclusions

The present study has shown that HRSV is the main cause of bronchiolitis in children, we also have Rhinovirus, and Bocavirus which also play a significant role. Data on the role played by SARS-CoV-2 in children with acute bronchiolitis is needed.

## Review registration

PROSPERO, CRD42018116067.

## Introduction

Bronchiolitis infection is included among the leading causes of hospitalization and death in pediatrics [1–5]. Bronchiolitis also represents a high economic burden on society and has generated a high hospital cost of around 72 million euros for children under the age of 2 in Portugal between 2000 and 2015 [3]. A study by Shi et al. found that lower respiratory infections due to Human Respiratory Syncytial Virus (HRSV), which is the main agent of bronchiolitis, causes approximately 3.2 million hospitalizations and about 60 thousand deaths per year worldwide in children under the age of 5 [6].

Bronchiolitis is generally considered to be a viral illness, and the most common viruses include HRSV, Influenza virus, Rhinovirus (RV), Human Metapneumovirus (HMPV), Enterovirus (EV), Human Coronavirus (HCoV), Human Parainfluenza Virus (HPIV), Human Adenovirus (HAdV), and Human Bocavirus (HBoV).

During the last two decades, the increase in use of Polymerase Chain Reaction (PCR) assays for the detection of respiratory viruses has led to a reassessment of the role played by viruses such as RV in acute respiratory infections [7, 8]. These molecular detection assays have also revealed new respiratory viruses such as HMPV and HBoV, and some RV and HCOV species [9–13]. The role of respiratory viruses in bronchiolitis and more particularly the newly described agents has not been synthesized yet. The objective of this study was to report the detection rate of viral agents using PCR and the associated risk factors with bronchiolitis in children ≤ 2 years in the pre-COVID-19 pandemic era.

## Methods

### Study design

This systematic review was conducted according to the principles of the Centre for Reviews and Dissemination [14]. The methodological standards of the PRISMA declaration have been applied for this review (S1 Table in S1 File) [15]. The protocol for this review has been registered in the PROSPERO database under number CRD42018116067. This review reports previous published data and ethical clearance was not required.

### Electronic search

Pubmed and Global Index Medicus were searched with a combination of keywords related to common respiratory viruses and bronchiolitis. The search strategy applied in Pubmed is

available in S2 Table in S1 File. The search strategy in Pubmed has been adjusted to the Global Index Medicus database. The databases have been consulted since their creation until 06 February 2019 and updated on 16 August 2020.

## Manual search

The literature search was supplemented by a review of references of included articles and relevant reviews. We have gradually updated our search strategy iteratively by adding the specific keywords of the searches found manually. No language or geographical restrictions have been applied.

## Integration criteria

Observational studies (cohort, case-control, and cross-sectional) published in peer-reviewed journals were used if they reported the detection of respiratory viruses using PCR assays in children < 2 years with bronchiolitis. A wide range of definition of bronchiolitis has been used by the authors of the included articles. Case reports, review, and duplicated studies were excluded.

## Study selection

Two investigators (SK and AFM) independently preselected the identified articles on the basis of their titles and abstracts (if necessary). The relevance of eligible studies was assessed using predefined integration criteria.

## Data extraction

Data from the included studies were independently extracted by two authors (SK and FBSN) via a preconceived form. The information gathered included: the name of the author, the year of publication, the design of the study, the country, the WHO regions, the sampling method, the period of the study, the definition of bronchiolitis, the exclusion criteria, the sample type, the viral detection assay, age range, mean or median age, percentage of male gender, number of samples tested for each virus, number of positives for each virus, and the data of the evaluation of the study quality. All disagreements were resolved by discussion between two authors and an arbitration by a third author if need be.

## Assessing research quality

The risk of bias of the included studies was estimated as low (8–10), moderate (5–7), and high (0–4) risk of bias using the Hoy et al. assessment tool (S3 Table in S1 File) [16].

## Data synthesis

Forest plots, summary tables and a narrative summary were used to present the overall results. The calculation of the prevalence took into account the weight of each study. Each prevalence was estimated using a random-effect meta-analysis given the inseparable heterogeneity of observational studies. A dual arcsine transformation of Freeman-Turkey was used to stabilize the variances in the prevalence calculation. Subgroup analysis was performed to estimate prevalence variations by sample type used for viral detection, WHO region and some viral codetection status. The Cochran Q test and the $I^2$ statistic were used to measure heterogeneity between studies [17]. Analysis based on studies with a low risk of bias, children $\leq$ 1 year, hospitalized children, and cross sectional were used for sensitivity analyses. Visual inspection of a funnel plot and the Egger test were used to estimate the risk of publication bias [18]. A

prediction interval was provided for all meta-analyses to predict future study values. Values of p <0.05 were assimilated as statistically significant. The analyses were conducted using the meta package version 4.9–2 under the R version 3.5.1 software [19, 20]. The code "metaprop" was used for meta-analyses of prevalence.

# Results

## Study selection and characteristics

The literature search provided a total of 3777 articles and 154 duplicates were excluded. Selection based on titles and abstracts excluded 3370 irrelevant articles. We therefore examined 253 complete texts and excluded 203 for multiple reasons (Fig 1, S4 Table in S1 File). We finally stayed with 50 articles (51 studies) that met the inclusion criteria [21–70]. The different bronchiolitis case definitions are showed in the S5 Table in S1 File. Children were recruited in included studies between October 1999 and December 2017 (Table 1). The selected studies were published between 2002 and 2020. Almost all children included in these studies were recruited consecutively. Most studies were cross-sectional studies (44; 86.3%), published in the English language (44; 86.3), carried out in Europe (28; 54.9%), carried out on a continuous period (35; 68.6%). Most studies had a prospective recruitment (44; 86.3%), a clear bronchiolitis case definition (45; 88.2%), and low risk of bias (34; 66.7%). Hospitalized boys less than 2 years were predominant in included studies. Analyzed samples were mainly nasopharyngeal secretions (45; 88.2%).

## Prevalence and codetection rate of viral infections among children < 2 years with bronchiolitis

HRSV was largely the most commonly detected virus (59.2%; 95% CI [54.7; 63.6]). Other viruses included in descending order: RV (19.3%; 95% CI [16.7; 22.0]), HBoV (8.2%; 95% CI [5.7; 11.2]), HAdV (6.1%; 95% CI [4.4; 8.0]), HPIV (5.4%; 95% CI [3.8; 7.3]), HMPV (5.4%; 95% CI [4.4; 6.4]), Influenza (3.2%; 95% CI [2.2; 4.3]), HCoV (2.9%; 95% CI [2.0; 4.0]), and EV (2.9%; 95% CI [1.6; 4.5]) (Fig 2, S1 Fig in S1 File). HRSV was the predominant virus involved in multiple detections. The most codetections were HRSV + RV (7.1%, 95% CI [4.6; 9.9]) and HRSV + HBoV (4.5%, 95% CI [2.4; 7.3]) (S2 Fig in S1 File). We did not find any major change in our results when we conducted sensitivity analyses that included only children < 1 year, hospitalized children, studies with bronchiolitis case definition, cross sectional studies, and low risk of bias studies (S6 Table in S1 File). Substantial heterogeneity was detected in overall prevalence and sensitivity analyses for all viruses. Publication bias was detected for HMPV and Influenza meta-analyses (S6 Table in S1 File and S3-S11 Fig in S1 File).

## Subgroup meta-analysis

The subgroup analysis showed a significant difference in the prevalence of HRSV (p < 0.001), RV (p < 0.001), HBoV (p < 0.001), HAdV (p < 0.001), HPIV (p < 0.001), mild HCoV (p < 0.001), and EV (p < 0.001) according to the WHO regions (S7 Table in S1 File). Lower prevalence was observed in Eastern Mediterranean for HRSV, HBoV, and EV; in America for RV and HAdV; and in Europe for HPIV. A significant increase in the prevalence was observed for continuous study period for HMPV (p = 0.005), HPIV (p = 0.005), and Influenza (p = 0.014). The subgroup analysis according to the type of sample revealed a significant increase in nasopharyngeal secretions for HRSV (p < 0.001).

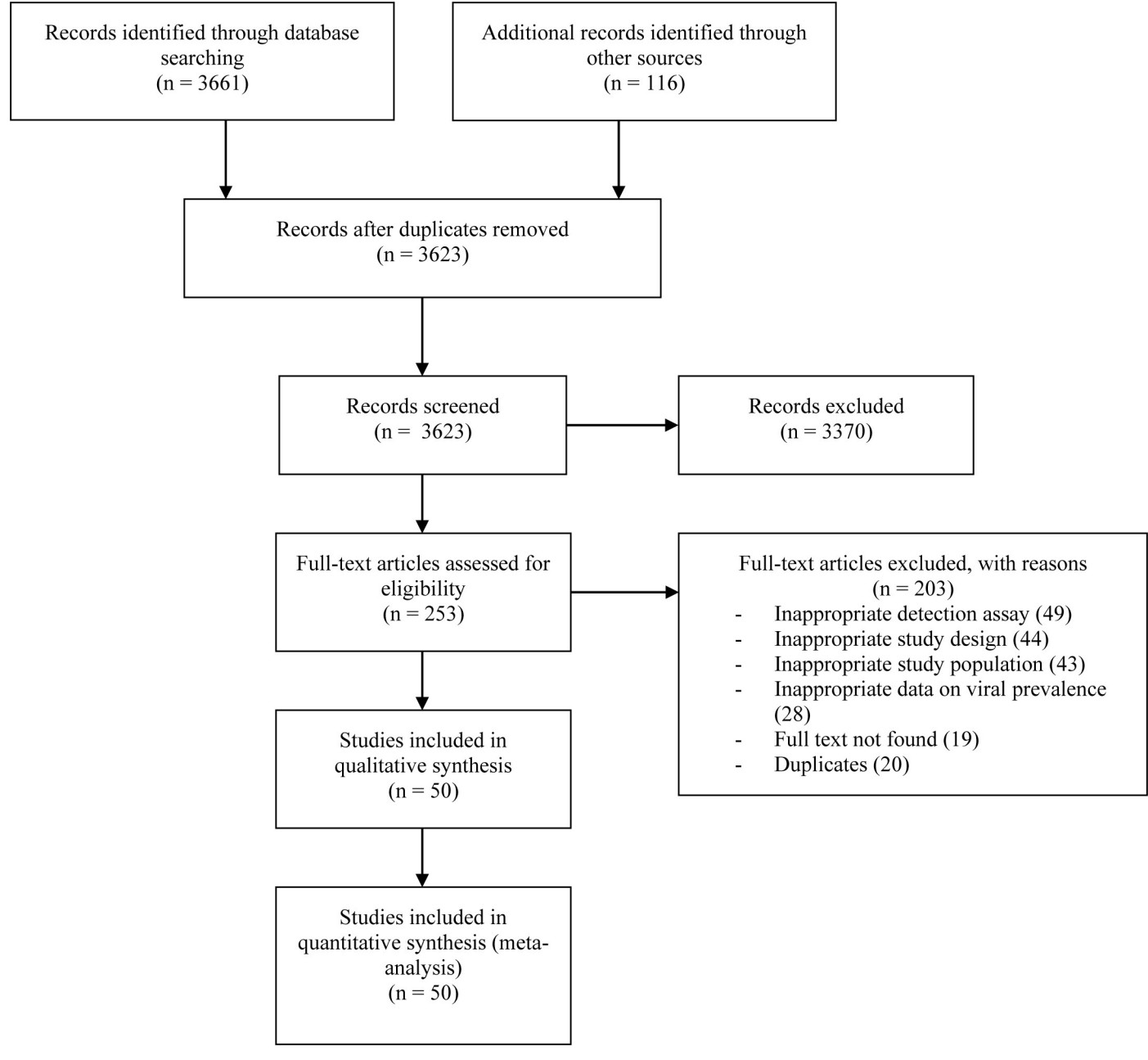

**Fig 1. Flow chart of research and selection of studies.**

## Discussion

This systematic review asserts the strong predominance of HRSV in children < 2 years with bronchiolitis. We have shown that HRSV is present in almost two-thirds of bronchiolitis cases. RV and two recently described viruses by the advent of molecular assays (HBoV and HMPV) were among the most common viruses. The HRSV + RV and HRSV + HBoV co-detections were the most frequent.

The predominance pattern of respiratory viruses in bronchiolitis reported in this study is consistent with that reported by several previous narrative reviews [71–73]. Regardless of

**Table 1. Sociodemographic and clinical characteristics of included studies.**

| Characteristics | N = 51 | % |
| --- | --- | --- |
| %Male; range | 48.0–72.5 | |
| Age range | | |
| • < 0.5 year | 2 | 3.9 |
| • < 1 Year | 17 | 33.3 |
| • < 1.5 years | 3 | 5.9 |
| • < 2 Years | 28 | 54.9 |
| • Unclear/Not reported | 1 | 2.0 |
| Period of inclusion of participants; range | Oct/1999-Dec/2017 | |
| Year of publication; range | 2002–2020 | |
| Study design | | |
| • Case control | 2 | 3.9 |
| • Cohort (Baseline data) | 5 | 9.8 |
| • Cross sectional | 44 | 86.3 |
| Sampling | | |
| • Consecutive sampling | 50 | 98.0 |
| • Simple random sampling | 1 | 2.0 |
| Timing of data collection | | |
| • Prospectively | 44 | 86.3 |
| • Retrospectively | 7 | 13.7 |
| Study bias | | |
| • Low risk of bias | 34 | 66.7 |
| • Moderate risk of bias | 17 | 33.3 |
| Seasonality | | |
| • Continuous study period | 35 | 68.6 |
| • Interrupted time series study | 15 | 29.4 |
| • Unclear/Not reported | 1 | 2.0 |
| WHO region | | |
| • America | 12 | 23.5 |
| • Eastern Mediterranean | 3 | 5.8 |
| • Europe | 28 | 54.9 |
| • South-East Asia | 1 | 1.9 |
| • Western Pacific | 7 | 13.7 |
| Language | | |
| • English | 44 | 86.3 |
| • Non English | 7 | 13.7 |
| Hospitalization | | |
| • Hospitalized | 40 | 78.4 |
| • Hospitalized/Outpatients | 4 | 7.8 |
| • Outpatients | 5 | 9.8 |
| • Unclear/Not reported | 2 | 3.9 |
| Bronchiolitis definition | | |
| • No | 6 | 11.8 |
| • Yes | 45 | 88.2 |
| Sample type | | |
| • Nasal secretions | 5 | 9.8 |
| • Nasopharyngeal secretions | 45 | 88.2 |
| • Throat secretions | 1 | 2.0 |

| Study | Total | | Prevalence (%) | 95% CI |
|---|---|---|---|---|

**HRSV (45 studies)**
**Random effect meta–analysis** **15351** **59.17** **[54.66; 63.60]**
Heterogeneity: $I^2$ = 96.8% [96.2%; 97.2%], $\tau^2$ = 0.0224, $p$ < 0.0001

**RV (36 studies)**
**Random effect meta–analysis** **12967** **19.29** **[16.67; 22.04]**
Heterogeneity: $I^2$ = 92.8% [90.9%; 94.2%], $\tau^2$ = 0.0092, $p$ < 0.0001

**HBoV (24 studies)**
**Random effect meta–analysis** **8706** **8.23** **[ 5.65; 11.24]**
Heterogeneity: $I^2$ = 95.4% [94.1%; 96.4%], $\tau^2$ = 0.0146, $p$ < 0.0001

**HAdV (26 studies)**
**Random effect meta–analysis** **6734** **6.08** **[ 4.37;  8.03]**
Heterogeneity: $I^2$ = 88.9% [85.0%; 91.8%], $\tau^2$ = 0.0079, $p$ < 0.0001

**HPIV (28 studies)**
**Random effect meta–analysis** **7933** **5.39** **[ 3.78;  7.26]**
Heterogeneity: $I^2$ = 91% [88.1%; 93.2%], $\tau^2$ = 0.0090, $p$ < 0.0001

**HMPV (32 studies)**
**Random effect meta–analysis** **9908** **5.38** **[ 4.40;  6.44]**
Heterogeneity: $I^2$ = 76.9% [67.7%; 83.5%], $\tau^2$ = 0.0027, $p$ < 0.0001

**Influenza (24 studies)**
**Random effect meta–analysis** **6571** **3.17** **[ 2.17;  4.34]**
Heterogeneity: $I^2$ = 82.1% [74.3%; 87.5%], $\tau^2$ = 0.0042, $p$ < 0.0001

**HCoV (27 studies)**
**Random effect meta–analysis** **7431** **2.91** **[ 1.96;  4.03]**
Heterogeneity: $I^2$ = 83.5% [76.9%; 88.1%], $\tau^2$ = 0.0047, $p$ < 0.0001

**EV (15 studies)**
**Random effect meta–analysis** **4202** **2.86** **[ 1.55;  4.52]**
Heterogeneity: $I^2$ = 86.1% [78.6%; 90.9%], $\tau^2$ = 0.0057, $p$ < 0.0001

**Overall random effect meta–analysis  79803** **13.26** **[10.98; 15.71]**
Residual heterogeneity: $I^2$ = 92.7% [92.0%; 93.3%], $p$ = 0

 0 20 40 60 80

**Fig 2. Global prevalence of respiratory viruses in children with bronchiolitis.**

multiple factors including detection assays, tested sample types, children's age, and infection severity most studies report that HRSV is the major agent in cases of bronchiolitis with rates ranging from 50 to 80% [72]. RV, that is the second most common virus in this study, has long

been considered a cause of benign respiratory infection such as the common cold [74]. Early investigations of the prevalence of RV in bronchiolitis episodes were conducted using traditional assays including serological testing and culture [75, 76]. The development of serological assays has always been difficult for RV because of the high number of serotypes. It is also well known that RV is insensitive to most cell lines used in viral isolation [77, 78]. Therefore the recent widespread use of PCR could justify this hidden importance of RV in low respiratory tract infections [79–84]. Many other recent works have also highlighted the importance of RV in other respiratory infections such as asthma, wheezing, and long-term respiratory sequelae in pre-school children [85–90].

HMPV and HBoV were first described in 2001 and 2005 respectively using molecular assays [9, 12]. As previously reported, our study further reinforces the role assigned to these relatively new agents in bronchiolitis [71–73]. Human Bocavirus is commonly reported in asymptomatic children and in co-detection with other viruses, which has long raised the question of its exclusive involvement in the pathogenesis of respiratory infections [91–96]. On the other hand, reports have also highlighted HBoV involvement in life-threatening respiratory illness in children, but this still does not establish a role for bocavirus as an important pathogen [97–99]. The HBoV reported in this study as the third most common virus in children with bronchiolitis therefore deserves further investigation to explain its clinical relevance in bronchiolitis.

This study has shown an increased codetection of either RV or HBoV in children infected with HRSV. This high prevalence of RV or HBoV in HRSV positive patients may indicate an overlap in the circulation period of these predominant viruses in bronchiolitis. It is known that HBoV and RV are recorded throughout the year, which obviously coincides with the circulation period of HRSV with peaks in winter and early spring [100, 101].

Respiratory viruses are characterized by their easy transmission through the contact of contaminated objects and through the airway [102–104]. These viruses are thus cosmopolitan and know no boundaries in their distribution across the different regions of the world. The frequencies of detection of respiratory viruses throughout the various regions of the world are governed by multiple climatic, sociodemographic and cultural factors. We are therefore unable to interpret the pattern of dominance observed in the prevalence of respiratory viruses according to the WHO regions observed during this work. The majority of interrupted time series studies included in this review were conducted in the HRSV peak circulation period [105]. Contrary to our expectations, the prevalence of HRSV in these interrupted time series studies was not significantly greater than studies conducted over a continuous period. On the other hand, the HMPV, the HPIV and Influenza presented a significant higher prevalence in studies performed in continuous period. This result suggests that the peaks of circulation of HMPV, HPIV and Influenza are different from that of HRSV. In fact the frequency of detection of respiratory viruses can vary from year to another. Some studies have shown an overlap of HRSV and HMPV circulation while others have not [106, 107]. This disagreement observed in the seasonal pattern of these respiratory viruses circulation compromises our ability to interpret our subgroup analysis according to the continuous or interrupted study period.

It is generally recognized that bronchiolitis is a respiratory condition of children up to the age of 2 years [108]. Several reports have shown that only clinical symptoms are enough for the management of children with bronchiolitis, and viral testing is generally considered unnecessary [109, 110]. Data on the impact of multiple viral infections on the severity of bronchiolitis is controversial and does not justify recommendations of routinely virological examinations in most guidelines [111, 112]. Increase use of healthcare resources for virus detection and regularly recorded codetections do not warrant cohorting of infected children and is another reason that hinder the investigation of viral pathogens in bronchiolitis clinical practice [110].

However, virological tests are still used in children with severe bronchiolitis to reduce the unnecessary use of antibiotics and interventions such as chest X-rays [108, 113]. In children admitted to receive palivizumab, American Academy of Pediatrics recommends HRSV testing for cessation of treatment if the test is positive [114].

Assessing the respective contribution of respiratory viruses in bronchiolitis could also be of crucial importance in orienting priority actions such as the allocation of financial resources for research on the main contributors, HRSV, particularly with the imminent introduction of vaccines against HRSV [115].

It is important to mention that the small number of studies in some subgroups analyses restricts our ability to draw definitive conclusions. A second limitation in our study is that we did not consider multiple other factors that could further explain the variability in the prevalence of viruses in bronchiolitis such as comorbidities, anti-HRSV prophylaxis and the number of virus types sought in studies for multi-species such as mild HCoV and HPIV. It is also known that bronchiolitis case definitions show great variability in terms of age limit and constellation of clinical symptoms according to geographic area and time that we did not consider in this study [71, 116].

Beyond these limitations, this systematic review and meta-analysis reports the prevalence recorded over two decades of a large panel of common respiratory viruses currently involved in bronchiolitis. We report the data obtained in almost all WHO regions from the PCR that currently represent the most commonly used assays in diagnosing respiratory viruses, which is another major asset of this work. We also have conducted multiple sensitivity analyses that further strengthen the robustness of our results on multiple important aspects such as children hospitalization, age range, and design and quality of studies.

The results of this systematic review and meta-analysis further underline the strong HRSV predominance in children with bronchiolitis. This work also highlights the importance of RV and the newly described HMPV and HBoV in children with bronchiolitis. The HRSV + RV and HRSV + HBoV co-detections were the most frequent in children with bronchiolitis.

The high costs of prophylaxis with palivizumab remain a major wall to its widespread use in reducing the burden of bronchiolitis in children. Therefore, to significantly reduce the heavy burden of bronchiolitis due to HRSV in children, the finalization and availability of HRSV vaccines is a high priority. Future studies should explore the involvement of multiple viral infections in the bronchiolitis severity and to clarify the clinical impact of HBoV in bronchiolitis cases. Social distancing measures undertaken for the SARS-CoV-2 response have shown a significant reduction in child hospitalizations due to acute bronchiolitis between the pre-pandemic and pandemic eras [117]. Although showing an association between the viral load and the severity of acute bronchiolitis, a recent study showed a relatively small contribution of the 4 HCoV species associated with mild diseases and most often associated with another coinfecting virus in a cohort of children with acute bronchiolitis in the pre-COVID-19 pandemic era [118]. To the best of our knowledge, no study has reported SARS-CoV-2 in a group of children with acute bronchiolitis to date. A few case reports of SARS-CoV-2 in children with acute bronchiolitis have been presented without exhaustive exploration of other common respiratory viruses [119, 120]. SARS-CoV-2 experimental infection in ferrets showed lung lesions consistent with acute bronchiolitis [121]. Collectively, these data suggest that involvement of SARS-CoV-2 solely in the development of acute bronchiolitis in children without comorbidities remains a matter of debate. More data on the role played by SARS-CoV-2 in children with acute bronchiolitis is needed.

## Supporting information

**S1 Checklist. PRISMA 2009 checklist.**
(PDF)

**S1 File.**
(ZIP)

## Author Contributions

**Conceptualization:** Sebastien Kenmoe, Richard Njouom.

**Data curation:** Sebastien Kenmoe, Cyprien Kengne-Nde, Jean Thierry Ebogo-Belobo, Donatien Serge Mbaga, Abdou Fatawou Modiyinji.

**Formal analysis:** Sebastien Kenmoe, Cyprien Kengne-Nde.

**Investigation:** Sebastien Kenmoe.

**Methodology:** Sebastien Kenmoe, Cyprien Kengne-Nde, Jean Thierry Ebogo-Belobo, Donatien Serge Mbaga, Abdou Fatawou Modiyinji.

**Project administration:** Sebastien Kenmoe, Richard Njouom.

**Supervision:** Sebastien Kenmoe, Richard Njouom.

**Validation:** Sebastien Kenmoe, Cyprien Kengne-Nde, Jean Thierry Ebogo-Belobo, Donatien Serge Mbaga, Abdou Fatawou Modiyinji, Richard Njouom.

**Writing – original draft:** Sebastien Kenmoe.

**Writing – review & editing:** Sebastien Kenmoe, Cyprien Kengne-Nde, Jean Thierry Ebogo-Belobo, Donatien Serge Mbaga, Abdou Fatawou Modiyinji, Richard Njouom.

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
