## [Decision Letter · Decision Letter 0]

16 Oct 2020

PONE-D-20-26375

Systematic review and meta-analysis of the prevalence of common respiratory viruses in children < 2 years with bronchiolitis reveal a weak role played by the SARS-CoV-2

PLOS ONE

Dear Dr. Njouom,

Thank you for submitting your manuscript to PLOS ONE. After careful consideration, we feel that it has merit but does not fully meet PLOS ONE’s publication criteria as it currently stands. Therefore, we invite you to submit a revised version of the manuscript that addresses the points raised during the review process.

Notably, it is especially important to carefully address the comment of reviewer 2, who flagged that the studies reviewed in your manuscript had included their patients before the start of the COVID-19 pandemic. Therefore, I agree with the reviewer that it is crucial to remove all reference to SARS-CoV-2 from the title, abstract and conclusions. 

We look forward to receiving your revised manuscript.

Kind regards,

Rik L. de Swart, Ph.D.

Academic Editor

PLOS ONE

Journal Requirements:

2.Thank you for stating the following financial disclosure:

 [The funders had no role in study design, data collection and analysis, decision to publish, or preparation of the manuscript.].

Reviewers' comments:

Reviewer's Responses to Questions

**Comments to the Author**

1. Is the manuscript technically sound, and do the data support the conclusions?

Reviewer #1: Yes

Reviewer #2: Partly

Reviewer #3: Partly

2. Has the statistical analysis been performed appropriately and rigorously? 

Reviewer #1: I Don't Know

Reviewer #2: Yes

Reviewer #3: Yes

3. Have the authors made all data underlying the findings in their manuscript fully available?

Reviewer #1: Yes

Reviewer #2: Yes

Reviewer #3: Yes

4. Is the manuscript presented in an intelligible fashion and written in standard English?

Reviewer #1: Yes

Reviewer #2: No

Reviewer #3: Yes

5. Review Comments to the Author

Reviewer #1: This is a reasonably well conducted systematic review which highlights COVID19 as not being a virus that causes bronchiolitis. I have minor suggestions to improve readability:

1. introduction section: "causes" not "cause" - Bronchiolitis infection is included among the leading causeS...

2. page 14: "reports have also highlighted HBoV involvement in life-threatening in children with severe respiratory infections" needs grammatical correction - for example it could be changed to "reports have also highlighted HBoV involvement in life-threatening respiratory illness in children". An additional reference to this section could be 10.1111/jpc.14587 , which describes the case of a young boy with life-threatening plastic bronchitis and only bocavirus on PCR testing as a potential cause.

Reviewer #2: Dear authors,

I have reviewed the manuscript (PONE-D-20-26375) entitled “Systematic review and meta-analysis of the prevalence of common respiratory viruses in children < 2 years with bronchiolitis reveal a weak role played by the SARS-CoV-2”. This is a Systematic review and meta-analysis study conducted on 51 studies in patients with Bronchiolitis. Although it doesn’t add something new to the literature, I believe that such studies in different periods from different regions are useble for clinical approaches to reflect that regions characteristics of the diseases. There are some issues that might improve the manuscript which are as follows;

1. Although the English style is not so bad, the languge should be checked by a native English speaker.

2. There has been added a little information at the part of “Discussion” and has not given newer information about bronchiolitis and lower respiratory tract infection with COVID-19. Some information should be added and The discussion part can be maked more attractive part for the readers. During reading the article; it should not to be boring.

Reviewer #3: The authors have carried out a rigorous literature search and collected data on viral testing for bronchiolitis in young children over 2 decades. However, it is markedly of note that all these children are being referenced as recruited between 1999-2017. The SARS-COV-2 was only describe din 2019 and testing became available in 2020. So it is unclear how any reference to SARS-COV-2 is being made at all in the paper?

If indeed I am not misunderstanding the data being presented, and all the children discussed were enrolled between 1999-2017, then there is gross overstatement of the name of SARS-COV-2 in this paper. It is mentioned in the title, in the abstract and in the results. "weak role played by SARS-COV-2" "no study reported presence of SARS-COV-2" are all misleading statements if none of the studies actually enrolled any patients since the pandemic and the novel SARS-COV-2 has been described.

This paper would be more acceptable if it was discussed for what it actually was, a review of common respiratory viruses in children under 2 with bronchiolitis in the pre-pandemic era. To that end, indeed a lot of work has been applied and that is the only work that should be discussed. Any unnecessary references to SARS-COV-2 or conclusions/results thereof should be removed.

- Please remove SARS-COV-2 from titles and from abstract since your study cannot discuss it at all.

Other language errors requiring correction

- Line 3: "among the leading causes..."

Line 10: reword "the most common... ?virus ?pathogen....include"

Lines 18-3-: Please remove all reference to SARS-COV-2 or explain better why your results, with patients last enrolled in 2017 have any relationship to the pandemic of 2019-20.

- Line 71-72: "data of the evaluation of the study quality"? unclear meaning, please reword.

- Line 160-161: unclear what the sentence " Therefore the recent widespread us of PCR....." means

-Line 170-171: have also highlightes HbOvinvolvement in life threatening in children....?" unclear menaing? missing words?

- Line 205-207 sentence starting with "Increase us eof health resources..." unclear meaning, seems either rincomplete or confused. Please clarify or reword.

- Line 212-214 : again unclear menaing, what conclusion do the authors wnat to draw here?

-Line 216: one drawback is the very small number of children actually being discussed in the studies that eventually met criteria. Its only 51 globally, over 20 years, for a condition as ocmmon as bronchiolitis and these common viral pathogens. So consluions are indeed hard to draw form such a small sample.

- Line 233-234: language incorrect please reword.

Line 241-244: Unclear meaning...are the authors suggesting that studies need ot be conducted on viral coinfections and their effect on the severity of bronchiolitis in order to reduce the use of viral testing in children with bronchiolitis?

6. PLOS authors have the option to publish the peer review history of their article (what does this mean?). If published, this will include your full peer review and any attached files.

Reviewer #1: No

Reviewer #2: No

Reviewer #3: No

---

## [Author Response · Author response to Decision Letter 0]

20 Oct 2020

Review Comments to the Author

Reviewer #1: This is a reasonably well conducted systematic review which highlights COVID19 as not being a virus that causes bronchiolitis. I have minor suggestions to improve readability:

Authors: We thank the reviewer for this appreciation. 

1. introduction section: "causes" not "cause" - Bronchiolitis infection is included among the leading causeS...

Authors: We thank the reviewer for this comment, the text is now modified accordingly.

2. page 14: "reports have also highlighted HBoV involvement in life-threatening in children with severe respiratory infections" needs grammatical correction - for example it could be changed to "reports have also highlighted HBoV involvement in life-threatening respiratory illness in children". An additional reference to this section could be 10.1111/jpc.14587 , which describes the case of a young boy with life-threatening plastic bronchitis and only bocavirus on PCR testing as a potential cause.

Authors: We thank the reviewer for this comment, the text is now modified accordingly.

Reviewer #2: Dear authors,

I have reviewed the manuscript (PONE-D-20-26375) entitled “Systematic review and meta-analysis of the prevalence of common respiratory viruses in children < 2 years with bronchiolitis reveal a weak role played by the SARS-CoV-2”. This is a Systematic review and meta-analysis study conducted on 51 studies in patients with Bronchiolitis. Although it doesn’t add something new to the literature, I believe that such studies in different periods from different regions are useble for clinical approaches to reflect that regions characteristics of the diseases. There are some issues that might improve the manuscript which are as follows;

1. Although the English style is not so bad, the languge should be checked by a native English speaker.

Authors: The manuscript has been carefully proofread and edited by a native English speaker. Thanks for the suggestion.

2. There has been added a little information at the part of “Discussion” and has not given newer information about bronchiolitis and lower respiratory tract infection with COVID-19. Some information should be added and The discussion part can be maked more attractive part for the readers. During reading the article; it should not to be boring.

Authors: We really appreciate the suggestion. The following paragraph has been added in discussion section. 

“Social distancing measures undertaken for the SARS-CoV-2 response have shown a significant reduction in child hospitalizations due to acute bronchiolitis between the pre-pandemic and pandemic eras 99. Although showing an association between the viral load and the severity of acute bronchiolitis, a recent study showed a relatively small contribution of the 4 HCoV species associated with mild diseases and most often associated with another coinfecting virus in a cohort of children with acute bronchiolitis in the pre-COVID-19 era 100. To the best of our knowledge, no study has reported SARS-CoV-2 in a group of children with acute bronchiolitis to date. A few case reports of SARS-CoV-2 in children with acute bronchiolitis have been presented without exhaustive exploration of other common respiratory viruses 101,102. SARS-CoV-2 experimental infection in ferrets showed lung lesions consistent with acute bronchiolitis 103. Collectively, these data suggest that involvement of SARS-CoV-2 solely in the development of acute bronchiolitis in children without comorbidities remains a matter of debate.”

Reviewer #3: The authors have carried out a rigorous literature search and collected data on viral testing for bronchiolitis in young children over 2 decades. However, it is markedly of note that all these children are being referenced as recruited between 1999-2017. The SARS-COV-2 was only describe din 2019 and testing became available in 2020. So it is unclear how any reference to SARS-COV-2 is being made at all in the paper?

Authors: Dear reviewer, thank you for this important comment. The search for articles in this manuscript was carried out until August 2020, during the SARS-CoV-2 pandemic period. However, as of this date, no study had yet reported the presence of SARS-CoV-2 in children with acute bronchiolitis. To the best of our knowledge, even to date, apart from a few case reports of children with acute bronchiolitis positive for SARS-CoV-2, no large studies have reported SARS-CoV-2 in a group of children with acute bronchiolitis. Collectively, all of the above explanations justify the involvement of SARS-CoV-2 in this work. We have also added the paragraph below in the discussion to support the interpretation of our findings related to SARS-CoV-2. 

“Social distancing measures undertaken for the SARS-CoV-2 response have shown a significant reduction in child hospitalizations due to acute bronchiolitis between the pre-pandemic and pandemic eras 99. Although showing an association between the viral load and the severity of acute bronchiolitis, a recent study showed a relatively small contribution of the 4 HCoV species associated with mild diseases and most often associated with another coinfecting virus in a cohort of children with acute bronchiolitis in the pre-COVID-19 era 100. To the best of our knowledge, no study has reported SARS-CoV-2 in a group of children with acute bronchiolitis to date. A few case reports of SARS-CoV-2 in children with acute bronchiolitis have been presented without exhaustive exploration of other common respiratory viruses 101,102. SARS-CoV-2 experimental infection in ferrets showed lung lesions consistent with acute bronchiolitis 103. Collectively, these data suggest that involvement of SARS-CoV-2 solely in the development of acute bronchiolitis in children without comorbidities remains a matter of debate.”

If indeed I am not misunderstanding the data being presented, and all the children discussed were enrolled between 1999-2017, then there is gross overstatement of the name of SARS-COV-2 in this paper. It is mentioned in the title, in the abstract and in the results. "weak role played by SARS-COV-2" "no study reported presence of SARS-COV-2" are all misleading statements if none of the studies actually enrolled any patients since the pandemic and the novel SARS-COV-2 has been described.

Authors: Dear reviewer, thank you for this important comment. We totally agree with the reviewer's comment and have altered the context of SARS-CoV-2 in the title and abstract of the article as shown below. 

“Title: Systematic review and meta-analysis of the prevalence of common respiratory viruses in children < 2 years with bronchiolitis reveal a paucity of data related to SARS-CoV-2”

“Abstract: More data on the role played by SARS-CoV-2 in children with acute bronchiolitis is needed.”

This paper would be more acceptable if it was discussed for what it actually was, a review of common respiratory viruses in children under 2 with bronchiolitis in the pre-pandemic era. To that end, indeed a lot of work has been applied and that is the only work that should be discussed. Any unnecessary references to SARS-COV-2 or conclusions/results thereof should be removed.

- Please remove SARS-COV-2 from titles and from abstract since your study cannot discuss it at all.

Dear Reviewer, we agree and have changed our original over-interpretation of SARS-CoV-2 for this article. However, the article search period overlaps very well with the SARS-CoV-2 pandemic era (search until August 2020). Given the current pandemic context, we maintain a focus on SARS-CoV-2 in the article as it emerges a need for SARS-CoV-2 investigations in children with acute bronchiolitis. We have also added a summary of the current SARS-CoV-2 situation in children with acute bronchiolitis.

Other language errors requiring correction

- Line 3: "among the leading causes..."

Authors: We thank the reviewer for this comment, the text is now modified accordingly.

Line 10: reword "the most common... ?virus ?pathogen....include"

Authors: We thank the reviewer for this comment, the text is now modified accordingly.

Lines 18-3-: Please remove all reference to SARS-COV-2 or explain better why your results, with patients last enrolled in 2017 have any relationship to the pandemic of 2019-20.

Dear Reviewer, the article search period overlaps very well with the SARS-CoV-2 pandemic era (search until August 2020). Given the current pandemic context, we maintain a focus on SARS-CoV-2 in the article as it emerges a need for SARS-CoV-2 investigations in children with acute bronchiolitis. We have also added a summary of the current SARS-CoV-2 situation in children with acute bronchiolitis.

- Line 71-72: "data of the evaluation of the study quality"? unclear meaning, please reword.

Authors: We thank the reviewer for this comment, the text is now modified accordingly.

- Line 160-161: unclear what the sentence " Therefore the recent widespread us of PCR....." means

Authors: We thank the reviewer for this comment, the text is now modified accordingly as shown below.

“Therefore, the recent widespread use of PCR could justify this hidden importance of RV in low respiratory tract infections.”

-Line 170-171: have also highlightes HbOvinvolvement in life threatening in children....?" unclear menaing? missing words?

Authors: We thank the reviewer for this comment, the sentence is now reworded accordingly as shown below.

“On the other hand, reports have also highlighted HBoV involvement in life-threatening respiratory illness in children, but this still does not establish a role for bocavirus as an important pathogen.” 

- Line 205-207 sentence starting with "Increase us eof health resources..." unclear meaning, seems either rincomplete or confused. Please clarify or reword.

Authors: We thank the reviewer for this comment, the sentence is now reworded accordingly as shown below.

“Increase use of healthcare resources for virus detection and regularly recorded codetections do not warrant cohorting of infected children and is another reason that hinder the investigation of viral pathogens in bronchiolitis clinical practice.”

- Line 212-214 : again unclear menaing, what conclusion do the authors wnat to draw here?

Authors: We thank the reviewer for this comment, the sentence is now reworded accordingly as shown below.

“Assessing the respective contribution of respiratory viruses in bronchiolitis could also be of crucial importance in orienting priority actions such as the allocation of financial resources for research on the main contributors, HRSV, particularly with the imminent introduction of vaccines against HRSV.” 

-Line 216: one drawback is the very small number of children actually being discussed in the studies that eventually met criteria. Its only 51 globally, over 20 years, for a condition as ocmmon as bronchiolitis and these common viral pathogens. So consluions are indeed hard to draw form such a small sample.

Dear Reviewer, although only 51 studies are included in the present work, we used strong and robust statistical methods and our conclusion is only based on our findings. Our study also includes high geographic (5 out of the 6 WHO regions represented) and temporal representativeness.

- Line 233-234: language incorrect please reword.

Authors: We thank the reviewer for this comment, the sentence is now reworded accordingly as shown below.

“The results of this systematic review and meta-analysis further underline the strong HRSV predominance in children with bronchiolitis.”

Line 241-244: Unclear meaning...are the authors suggesting that studies need ot be conducted on viral coinfections and their effect on the severity of bronchiolitis in order to reduce the use of viral testing in children with bronchiolitis?

Authors: We thank the reviewer for this comment, the sentence is now reworded accordingly as shown below.

Future studies should explore the involvement of multiple viral infections in the bronchiolitis severity and to clarify the clinical impact of HBoV in bronchiolitis cases. More data on the role played by SARS-CoV-2 in children with acute bronchiolitis is also needed.

---

## [Editor Report · Decision Letter 1]

28 Oct 2020

PONE-D-20-26375R1

Systematic review and meta-analysis of the prevalence of common respiratory viruses in children < 2 years with bronchiolitis reveal a paucity of data related to SARS-CoV-2

PLOS ONE

Dear Dr. Njouom,

Thank you for submitting your manuscript to PLOS ONE. After careful consideration, we feel that it has merit but does not fully meet PLOS ONE’s publication criteria as it currently stands. Therefore, we invite you to submit a revised version of the manuscript that addresses the points raised during the review process.

The revised manuscript has adequately addressed the majority of the comments of the reviewers. However, I do not agree with your response to the comment of reviewer 3, which was reiterated in my decision letter. The COVID-19 outbreak was declared a pandemic in March 2020, followed by several months of lockdowns and high pressure on global health research infrastructures. Therefore, the argument that the search for articles was continued until August 2020 does not justify interpretation of the results of this systematic review in the context of the COVID-19 pandemic. All available reports were based on pre-pandemic data.

PLOS ONE explicitly considers systematic review papers for publication, as described in the submission guidelines (https://journals.plos.org/plosone/s/submission-guidelines#loc-systematic-reviews-and-meta-analyses). In my opinion, this section is of crucial importance: “A systematic review paper, as defined by The Cochrane Collaboration, is a review of a clearly formulated question that uses explicit, systematic methods to identify, select, and critically appraise relevant research, and to collect and analyze data from the studies that are included in the review.” Based on this, I cannot accept your objective, as stated in the abstract: “This systematic review and meta-analysis was initiated to clarify the prevalence of respiratory viruses in children with bronchiolitis in the coronavirus disease 2019 pandemic context.” It will be impossible to perform a systematic review with that aim until well into next year. 

I therefore ask you to revise the manuscript and remove all reference to the COVID-19 pandemic from the title, abstract, introduction and results section of your manuscript. I believe that it would be justified to add a paragraph at the end of the discussion in which you discuss the potential impact of the COVID-19 pandemic on bronchiolitis, and the necessity to collect data on this topic. Moreover, please make sure that PRISMA flow diagram is included as Fig 1 and the PRISMA checklist as supporting information, as specified in the instructions for authors. Finally, make sure that your figure TIF files are correctly uploaded to the system, as the current PDF file reports file errors and does not display your figures.

We look forward to receiving your revised manuscript.

Kind regards,

Rik L. de Swart, Ph.D.

Academic Editor

PLOS ONE

---

## [Author Response · Author response to Decision Letter 1]

29 Oct 2020

Review Comments to the Author

Editor: The revised manuscript has adequately addressed the majority of the comments of the reviewers. However, I do not agree with your response to the comment of reviewer 3, which was reiterated in my decision letter. The COVID-19 outbreak was declared a pandemic in March 2020, followed by several months of lockdowns and high pressure on global health research infrastructures. Therefore, the argument that the search for articles was continued until August 2020 does not justify interpretation of the results of this systematic review in the context of the COVID-19 pandemic. All available reports were based on pre-pandemic data.

PLOS ONE explicitly considers systematic review papers for publication, as described in the submission guidelines (https://journals.plos.org/plosone/s/submission-guidelines#loc-systematic-reviews-and-meta-analyses). In my opinion, this section is of crucial importance: “A systematic review paper, as defined by The Cochrane Collaboration, is a review of a clearly formulated question that uses explicit, systematic methods to identify, select, and critically appraise relevant research, and to collect and analyze data from the studies that are included in the review.” Based on this, I cannot accept your objective, as stated in the abstract: “This systematic review and meta-analysis was initiated to clarify the prevalence of respiratory viruses in children with bronchiolitis in the coronavirus disease 2019 pandemic context.” It will be impossible to perform a systematic review with that aim until well into next year. 

I therefore ask you to revise the manuscript and remove all reference to the COVID-19 pandemic from the title, abstract, introduction and results section of your manuscript. I believe that it would be justified to add a paragraph at the end of the discussion in which you discuss the potential impact of the COVID-19 pandemic on bronchiolitis, and the necessity to collect data on this topic.

Authors: We thank the Editor for these suggestions, the text is now modified accordingly.

Moreover, please make sure that PRISMA flow diagram is included as Fig 1 and the PRISMA checklist as supporting information, as specified in the instructions for authors.

Authors: We thank the Editor for this suggestion, Fig 1 is now shown in the S1 table.

Finally, make sure that your figure TIF files are correctly uploaded to the system, as the current PDF file reports file errors and does not display your figures.

Authors: New figures in TIFF formats have been uploaded to the system, thank you.

---

## [Editor Report · Decision Letter 2]

2 Nov 2020

Systematic review and meta-analysis of the prevalence of common respiratory viruses in children < 2 years with bronchiolitis in the pre-COVID-19 pandemic era

PONE-D-20-26375R2

Dear Dr. Njouom,

We’re pleased to inform you that your manuscript has been judged scientifically suitable for publication and will be formally accepted for publication once it meets all outstanding technical requirements.

Kind regards,

Rik L. de Swart, Ph.D.

Academic Editor

PLOS ONE
---

## [Editor Report · Acceptance letter]

4 Nov 2020

PONE-D-20-26375R2 

Systematic review and meta-analysis of the prevalence of common respiratory viruses in children < 2 years with bronchiolitis in the pre-COVID-19 pandemic era 

Dear Dr. Njouom:

I'm pleased to inform you that your manuscript has been deemed suitable for publication in PLOS ONE. Congratulations! Your manuscript is now with our production department. 

Kind regards, 

on behalf of

Dr. Rik L. de Swart 

Academic Editor

PLOS ONE